# Musings from the Tribbles Research and Innovation Network

**DOI:** 10.3390/cancers13184517

**Published:** 2021-09-08

**Authors:** Miriam Ruiz-Cantos, Claire E. Hutchison, Carol C. Shoulders

**Affiliations:** Centre for Endocrinology, William Harvey Research Institute, Queen Mary University of London, Charterhouse Square, London EC1M 6BQ 1, UK; m.ruizcantos@qmul.ac.uk (M.R.-C.); c.hutchison@qmul.ac.uk (C.E.H.)

**Keywords:** gene annotation, transcription, translation, GTEx

## Abstract

**Simple Summary:**

The Tribbles Research and Innovation Network (TRAIN) was developed in the context of the medical need to understand the contribution that Tribbles (TRIB) proteins make in regulating processes governing the physiological functioning of macrophages and other immune cells, adipocytes and prostate epithelial cells in immuno-metabolic disease (such as obesity) and several cancers. To summarize, TRAIN provided a cohort of PhD students with a unique opportunity to undertake multidisciplinary research aimed at uncovering the cell-specific mechanisms by which TRIBs exert control over immuno-metabolism and impact on prostate cancer progression. The associated training programme was enhanced by contributions from the TRIB community. This review highlights the science that motivated the establishment of TRAIN, including the discovery of how the TRIB1 protein enables the marking of the transcription factor C/EBPα for degradation, and regulatory mechanisms underlying cell-specific TRIB expression. It briefly reflects on the type of TRAIN-associated tools developed for future TRIB research.

**Abstract:**

This commentary integrates historical and modern findings that underpin our understanding of the cell-specific functions of the Tribbles (TRIB) proteins that bear on tumorigenesis. We touch on the initial discovery of roles played by mammalian TRIB proteins in a diverse range of cell-types and pathologies, for example, TRIB1 in regulatory T-cells, TRIB2 in acute myeloid leukaemia and TRIB3 in gliomas; the origins and diversity of *TRIB1* transcripts; microRNA-mediated (miRNA) regulation of *TRIB1* transcript decay and translation; the substantial conformational changes that ensue on binding of TRIB1 to the transcription factor C/EBPα; and the unique pocket formed by TRIB1 to sequester its C-terminal motif bearing a binding site for the E3 ubiquitin ligase COP1. Unashamedly, the narrative is relayed through the perspective of the Tribbles Research and Innovation Network, and its establishment, progress and future ambitions: the growth of TRIB and COP1 research to hasten discovery of their cell-specific contributions to health and obesity-related cancers.

## 1. Introduction

On reflection, my group’s interest in the TRIB proteins goes back some 12 years. Then, our question was quite simple and primarily motivated by the observation that variants at the *TRIB1* locus significantly increased an individual’s risk of developing high blood cholesterol and triglyceride levels, and coronary heart disease [1]. Today, as our appreciation and understanding of the work of others has increased, including in the field of cancer, the questions have proliferated and our fascination with this family of pseudokinases has grown. Here, we acknowledge some of the TRIB-research and people that have made it so and refer readers to some excellent review articles [2,3,4,5,6] that are interesting but tangential to the central theme of this commentary: why dissecting the cell-specific functions of TRIB isoforms could help cut through the biological complexities of obesity-related cancers. A particular focus, on *TRIB1* genomics, helps showcase the plethora of public resources now available for studying gene expression, while the inclusion of the reported TRIB1 protein structures highlights how the central kinase-like domain of the mammalian TRIB proteins may bind a specific yet diverse range of proteins. Notably, the structures also provide insights into how these pseudokinases bind the E3 ubiquitin ligase COP1 via a C-terminal motif to promote proteasomal-mediated degradation of TRIB-captured substrates. Additionally, we reflect on recent scientific advances and anticipated future developments that could shape the TRIB-research community, viewed through the prism of our (past) membership of the Tribbles Research And Innovation Network (i.e., TRAIN), the members of which are included in the Acknowledgement section of this commentary.

## 2. Establishment of TRAIN

In retrospect, it is clear that persistence and several pivotal events created TRAIN, the brainchild of Professor Endre Kiss-Toth. This persistence involved re-drafting a research and training programme with three main ambitions at its heart. First, to develop and share research tools for studying the complexities of the cell-specific activities of TRIB proteins in common potentially obesity-related diseases, including prostate [7] and breast [8] cancer. Second, to establish strong and sustainable collaborations embodying a broad knowledge base (and appropriate skill sets) that would underpin the interrogation of the growing list of suggested roles of TRIB isoforms in relation to immuno-metabolic disease and malignancy. Third, to provide a multi-disciplinary and inter-sectorial training programme to create a new generation of scientists with a strong springboard to take forward a ‘holistic’, integrated approach to developing TRIB-targeting therapeutics, including the design of small molecules that selectively modulate physical contacts between the central kinase-like domain of TRIB proteins with a specific protein or subset of proteins.

Arguably, the most direct evidence that the mammalian TRIB proteins might regulate cell-specific processes driving, or indeed suppressing, malignancy came from studies on flies and frogs. For example, genetic screens of *Drosophila* mutants clearly revealed that its single Tribbles gene (*Trbl)* regulates many steps in cell proliferation and migration [9], including those required for embryonic [10] and neural [11] development, bristle formation [12] and oogenesis [13]. Moreover, it was well-established that Trbl works by regulating the turnover of key proteins in a cell-specific manner, for example, in cells that form the mesoderm anlage, Trbl degrades the mitotic activator String/CDC25, delaying their cell-division until gastrulation is complete [9], whereas in border cells, which migrate to the anterior end of the oocyte to form a shell-structure through which sperm can enter, Trbl degrades Slbo [14], a key transcription factor promoting border cell migration [15,16].

Four discoveries hinting at the potential cell-specific roles of mammalian TRIB proteins in regulating a diverse range of cell signaling activities, and the ensuing collaborations arising from them, were pivotal in inspiring and creating the TRAIN. One, made by Professor Guillermo Velasco, identified strong association between TRIB3 expression and apoptosis in tumour cells [17]. In brief, Guillermo’s story started in the (and not a) cannabinoid field when his then-PhD student Arkaitz Carracedo analysed the transcriptomes of two subclones of glioma cells that displayed different sensitivities to Δ^9^-tetrahydrocannabinol (THC)-induced apoptosis. In the sensitive cell line, THC induced *TRIB3* expression, while in the THC-resistant cell line, enforced *TRIB3* expression rescued sensitivity to THC-induced apoptosis. Guillermo’s group, in collaboration with Endre, then extended the in vitro phenomenon to an in vivo system by showing that RasV^12^/E1A-*Trib3*-deficient murine tumour xenografts were also resistant to THC-induced autophagy-mediated cell death [18]. Their next advance, demonstrating that TRIB3-induction by THC inhibited the AKT/mTORC1 axis, corroborated earlier evidence that TRIB3 was likely to influence multiple metabolic processes, and vice versa. In fact, the first-ever sighting of TRIB3 came from a gene expression study that showed that *fld2* (i.e., *Trib3*) RNA was markedly raised in the livers of suckling *Lpin1*-deficient mice who had fatty liver, plus abnormalities of lipid and glucose metabolism, as a secondary manifestation of impaired adipose tissue development [19,20]. Similarly, *Trib3* RNA was discovered to be elevated in immortalised brown pre-adipocytes prepared from insulin receptor substrate-1 gene knock-out mice [21], characterised by a mild-to-moderate post-receptor insulin resistance phenotype, resembling that of non-insulin-dependent diabetes mellitus at the prediabetes stage.

Another aha moment was the discovery that TRIB3 binds acetyl-coenzyme A carboxylase (ACC) and the E3 ubiquitin ligase COP1 [22], which two years earlier had been shown to be a critical negative regulator of the tumour suppressor protein p53 [23]. In particular, Montminy and colleagues [22] demonstrated that full-length TRIB3 bound both ACC and COP1, whereas a truncated TRIB3 protein lacking the C-terminal 40 amino acid residues could only bind ACC. Compelling data from Karen Keeshan and Warren Pear [24] also revealed that TRIB2 contains separate binding sites for COP1 and its protein substrate, CCAAT/enhancer protein-α, and that deletion of the COP1 binding site abrogated TRIB2′s ability to degrade this transcription factor, block granulocytic differentiation and induce acute myelogenous leukaemia. In short, and more on this later, a common theme was emerging: TRIB proteins affect multiple cellular functions by binding a wide-range of proteins, including some that they recruit for COP1 to mark for degradation.

The third TRAIN-shaping discovery emerged from a high-throughput screening program designed to identify protein sequences mediating cytokine responses [25], the result of which could so easily have been disregarded. Thus, in outline, when Endre transfected cells with a library of cDNA sequences and assayed their ability to regulate the human *IL-8* promoter, he found one that suppressed IL-8 activity despite it having no open reading frame. It turned out that this unusual IL-8-suppressing construct contained a portion of the 3′UTR of *TRIB1*. Pursuing its mechanism of action, Endre then established that the UTR fragment increased endogenous *TRIB1* RNA levels in the transfected cells >10-fold and, to cut a long story short, repressed *IL-8* promotor activity by suppressing the activation of the Jun kinase AP-1, the putative transforming gene of avian sarcoma virus 17. Thus, he sought, and found, evidence that TRIB1 binds via its central kinase-like domain to specific mitogen-activated protein kinase (MAPK) and proposed that mammalian TRIB proteins may, by binding to MAPK, control MAPK activation and, therefore, a cell’s response to a diverse range of stimuli, including mitogens and pro-inflammatory cytokines [25,26].

The fourth discovery that inspired the scientific make-up of TRAIN sprang from Professor Sophie Brouard’s work on chronic antibody-mediated rejection (AMR) of grafts in kidney transplant recipients [27]. Unexpectedly, Sophie found that *TRIB1* RNA levels in their blood were a more specific and sensitive marker of AMR graft-failure than even *TRIB1* in their actual renal biopsy samples. Why this might be the case became (a bit) clearer upon finding that *TRIB1* is expressed in many peripheral white blood cells, importantly, in the CD4^+^CD25^high^CD127^−^Fox3^+^ subset of regulatory T cells [28,29] that suppress cellular immunity but at the expense of increasing the risk of cancer [30]. Pondering this result, alongside those produced by others in the field [31,32], had two consequences: appreciation of the enormity of the task that lay ahead in unravelling the isoform- and cell-specific pathogenic versus protective functions of TRIBs in circulating and tissue-resident immune cells, as well as in tumour and metabolic cells; and of the fact that progress on these fronts would be stimulated through a collaborative effort and establishment of a strong Tribbles community focused on this question. TRAIN was conceived, and its progress nurtured by the wider TRIB community, including Professors Robert Bauer, Leonard Dobens, Patrick Eyers, Zhuowei Hu, Wolfgang Link, Karen Keeshan, Peter Mace and Warren Pear.

## 3. Tissue- and Cell-Specific Expression of *TRIB* Family Members

Mammalian *TRIB1* and *TRIB2* transcripts were first identified in the thyroid gland [33,34], where *TRIB1* has a relatively high abundance (Figure 1). *TRIB3*, by contrast, was found to be relatively highly expressed in the liver [35], as judged by a Northern blot analysis. That said, it is also apparent from inspecting the Gene-Tissue Expression (GTEx) project data that *TRIB1-3* RNA levels exhibit a high degree of inter-person variability in many tissues [36], underlining the value of genome-wide RNA studies in large cohorts to obtain a more objective view of relative isoform-specific *TRIB* transcript numbers. In the GTEx survey of liver samples, for example, the median number of *TRIB1* transcripts per million (TPM) is 34.61, compared to TPMs of 22.1 and 59 for the 25th and 75th percentile points (Figure 1). Moreover, eighteen samples contained well in excess of 100 TPM. Similarly, in lung, there is a 2.4- and 2.7-fold difference between the 25th and 75th percentiles values for *TRIB1* and *TRIB3* RNA contents, respectively, compared to just a 1.5-fold difference for *TRIB2* RNA. It is also notable that there is a relatively tight distribution of *RFWD2* (encodes COP1) RNA levels in many bodily tissues, although in the liver there is a 1.8-fold difference between the 25th (7.2 TPM) and 75th (12.7 TPM) percentile values.

It would be tempting to ignore the relatively high inter-personal variability in tissue *TRIB* RNA levels due to the acknowledged technical limitations of the GTEx project [38,39,40,41], for example, differences in the cell compositions of tissues. However, to do so seems premature until we know more about the biology regulating *TRIB* expression in specific cell-types, and in co-culture systems that recapitulate the cellular and physiological complexities of tissues. Thus, we envisage that these data will identify specific confounding variables affecting *TRIB* expression, including in specific cell-types, and that these confounders will contribute to the observed variances of *TRIB* RNA levels in the GTEx tissues. It is also highly probable that genetic variants add to this variance by affecting *TRIB* transcript production and decay, and that many such variants will be discovered in further analyses that correct for the effects of non-genetic factors on *TRIB* expression.

Homing in on the study of *TRIB* expression in specific cell-types, we are especially drawn to the painstaking analysis of hemopoietic cells undertaken by Keeshan and colleagues [42]. First, the analysis itself, which nicely shows, through the collation of publicly available data, the expression profiles of *TRIB1-3* in murine and human hemopoietic stem, progenitor and mature cells, including in those of leukemic patients. Second, the biological inferences made by Karen from these analyses, for example, changes in *TRIB1* expression may contribute to the progression of HOX9-induced myeloid leukaemia, as subsequently shown [43], and TRIB3 may play a novel role in stem cell quiescence.

## 4. Genomic and Structural Data: Insights into TRIB Expression

The past five years have witnessed remarkable progress in understanding the contribution of the myriad of nuclear and cytoplasmic processes regulating transcript diversity and abundance [44,45,46,47,48]. Particularly helpful resources for this type of analysis include RNA-sequencing (RNA-seq) and Cap analysis of gene expression (CAGE) data [49] generated by the Encyclopaedia of DNA Elements (ENCODE) and Functional Annotation of Mammalian Genome consortiums. We have used CAGE to map the transcription start sites of *TRIB1* and have been reminded that annotated transcripts in the genome browsers may represent degradation products rather than a *bona fide* transcript with the usual features initiating translation: the 7-methyl-guanosine cap at the 5′ end and a 3′ polyadenylated (poly A) tail. To demonstrate this point, we compare and contrast genomic data for two *TRIB1* transcripts: *TRIB1*-201, designated on the Ensembl browser, as the principal species, and *TRIB1*-203 as a product of alternative splicing (Figure 2). The latter, which may derive from spurious transcription [50], lacks exon 1 sequences, has a short 3′UTR and no poly A tail, and is predicted by Ensembl to initiate translation from Met167.

From conventional cDNA studies and CAGE analyses, we infer that *TRIB1* transcription is predominantly initiated from multiple sites centred on a 220bp stretch of DNA ~550 upstream of the canonical translation initiation codon of TRIB1 in a cell-specific manner (Figure 3). Moreover, we infer that such transcripts are 5′-capped and exported to the cytoplasm (Figure 3) attached to both their cap-binding complex and exon-junction complexes (EJCs) assembled on exon-termini joined by splicing. Within the cytoplasm, the cap-binding complex is replaced by the eukaryotic translation initiation factor, thereby promoting recruitment of the 43S ribosomal pre-initiation complex to the m7-G Cap, whose main role is to locate and pin the translation initiation codon to further components of the translation machinery. For some *TRIB1*-201 transcripts, the pre-initiation complex would need to scan (in 5′ to 3′ direction) at least 927 nucleotides of the capped transcript before it finds the start codon (Figure 3), enabling TRIB1 protein synthesis to proceed.

Conventional cDNA cloning and sequencing methodologies and RNA-seq data (Figure 4A) now clearly show that the *TRIB1*-203 transcript reflects splicing at a cryptic donor splice-site deep within intron 1 (Figure 2 and Figure 4A). A re-appraisal of the protein-coding status and, thus, potential significance of this alternatively-spliced transcript is thus urgently needed to understand whether its expression might contribute, for example, to a cell’s response to a diverse range of stimuli, including mitogens and pro-inflammatory cytokines [25,26]. Currently, standard genome annotation methods indicate that *TRIB1*-203 uses a translation initiation codon within exon 2 (Met167) to generate a 206 amino acid protein (Figure 2). However, it seems implausible from structural data that this N-terminally truncated protein could adequately control ubiquitination of its target protein substrates (Figure 4B). As Peter Mace so eloquently explained at the second international symposium on Tribbles and Diseases [51], the αC-helix (amino acids 127–140) and β4 strand (amino acids 146–151) of TRIB1 (Figure 4B) protect it from COP1-mediated ubiquitination and ensuing proteasome-mediated degradation. In brief, these two structural elements sequester the C-terminal COP1-binding motif, preventing it from recruiting COP1. Protein substrates, such as C/EBPα, bind to a central portion of TRIB1’s pseudokinase domain (Figure 4C) and initiate changes that destabilise the COP1-motif binding pocket, thereby freeing the COP1-binding peptide to seek out COP1 and ubiquitinate the TRIB1-bound substrate. Hence, the potential regulatory importance of limiting the production of the putative *TRIB1*-203 gene-product is that it could bind COP1 unimpeded, initiate self-ubiquitination and degradation of this putative *TRIB1* isoform and adversely affect its ability to recruit a wide range of protein substrates to COP1 to facilitate their ubiquitination and subsequent degradation.

The finding that spuriously transcribed transcripts are degraded shortly after their transcription to limit their accumulation and potential for translation [50] may explain, at least in part, the low abundance of the *TRIB1*-203 transcript in a range of tissues (Figure 4A) and cell lines. As illustrated in Figure 5A, we find that capped *TRIB1*-203 transcripts are a very minor *TRIB1* species in the nucleus of ten commonly used cell lines, with RNA-seq data returning read profiles that one would expect to see for cleaved product (compare Figure 5B with Figure 5C). CAGE data further show that capped *TRIB1*-203 transcripts are rare in the cytoplasm (i.e., one or none per cell line), consistent with RNA-seq estimates of *TRIB1*-203 mRNA abundance in these cell lines (Figure 5D). Less clear is whether cytoplasmic RNA decay prevents cap-mediated translation of *TRIB1*-203 transcripts and, in particular, what role EJC-dependent nonsense mediated degradation [54,55] might play in determining the abundance and translation capacity of this and other alternatively spliced *TRIB* transcripts, including in malignancy.

As highlighted by the research of Niespolo and colleagues [56] on behalf of TRAIN, we now know of at least two microRNAs (miRNAs) that, through their binding to the 3′UTR of *TRIB1*, increase IL-8 expression. However, a full description (and even understanding) of how this class of RNAs regulate TRIB1 protein levels will require significant advances in miRNA technologies. These include the development of robust high-throughput assays to quantify the individual, combined and cooperative effects of 1000+ candidate miRNAs (Figure 6) on *TRIB1* transcript decay and translation, as well as software to compare expression values under basal and induced conditions. From data produced so far, it seems entirely feasible that *TRIB1*-targeting miRNAs have medium-to-large effect-sizes on *TRIB1*-transcript levels and translatability, and that their expression is regulated by both genetic and cell-specific signals. Moreover, as the number of *TRIB1*-binding miRNAs in a cell increase, this could have a more direct and larger impact on TRIB1-mediated functions than a cis-acting *TRIB1* disease-risk variant identified in large population type studies.

## 5. Growth of TRIB (and COP1) Research

There is no doubt that developing a website of available TRIB-research tools will help the TRIB community, but for now we are confined to just highlighting those in the TRAIN tool-box, some of which can be accessed via contact with a TRAIN consortium member (Acknowledgements). An important early decision was to use a standard set of prostate cancer cell lines (Table 1) in gene expression and cell-biological analyses, and in xenograft models of human prostate cancer. Notably, TRIB1 protein levels are relatively high in the PC3 cell line, whereas COP1 is undetectable, raising the question whether COP1 under specific stimulatory conditions may mark TRIB1 for proteasome-mediated degradation. Increasing TRIB1 expression in these PC3 cells has been shown to promote tumour growth [58], whereas increasing TRIB1 in DU145 cells did not [59]. Consequently, we keenly await proteomic analyses of the TRIB-interactome in the COP1-less PC3 cells, as well as structural studies that define the effects of TRIB1-captured protein substrates on the release of its sequestered C-terminal COP1-binding motif. Once these analyses have been undertaken, we may be better placed to understand the relative balance of TRIB’s different functions: providing a scaffold for signalling molecules to assemble versus acting as a COP1 recruiting sergeant for accelerating protein degradation. We are also very curious to understand under what circumstances TRIB1 (and potentially TRIB2, TRIB3) promotes nuclear COP1 retention [60], and to what effect?

Returning to the TRAIN tool-box, this includes a series of scripts to interrogate and visualise genomic data, plasmid and viral constructs, and murine models to study the physiological roles of cell-specific TRIB1 and TRIB3 expression, for example, in myeloid cells [70], prostate-specific *Pten*-deficient mice [59] and Tregs (*Trib1*^KO^, richard.danger@univ-nantes.fr; sophie.brouard@univ-nantes.fr). Finally, we would like to thank the wider TRIB community for their contribution to student TRAIN training events and bi-annual international TRIB conferences. Reflecting on the next TRIB conference, planned for the spring of 2023 under the auspices of Professors Robert Bauer, Leonard Dobens and Warren Pear, we look forward to meeting up in person and finding out how the collaborative approaches adopted by the TRIB community have accelerated mechanistic understanding into how each TRIB protein and binding partner contribute mechanistically to tumorigenesis.

## 6. Conclusions

TRIB proteins determine the abundance and subcellular distribution of a diverse range of proteins. An unknown proportion of TRIB-captured proteins are marked for degradation by the E3 ubiquitin ligase COP1, which binds to TRIB proteins via their C-terminal motifs. Importantly, COP1 has also been shown to regulate protein levels in a non-TRIB-mediated process, and there are data to suggest that TRIB1 inhibits nuclear export of COP1 once it enters the nucleus. However, data to compare how strongly endogenous TRIB proteins and non-TRIB protein complexes compete for recruitment of COP1 to promote proteasomal-mediated degradation of their captured proteins are lacking. In contrast, and as outlined in this review, publicly available genomic datasets are a valuable resource to better understand mechanisms of cell-specific regulation of mammalian TRIB and COP1 (i.e., *RFWD2*) expression. Now, we await the results of experimental exploration of how altered TRIB/COP1 expression affects isoform-specific TRIB abundance, subcellular distribution and downstream functions—a pre-requisite for designing a range of TRIB/COP1-therapeutics to ameliorate obesity-related cancers. Regarding drug development, we envisage that the diverse, and potentially opposing, biological activities of TRIB proteins and COP1 may limit their consideration as specific targets, and that a more promising option will be the development of small molecules that suppress/induce TRIB/COP1 binding to a subset of their downstream effectors.

## Figures and Tables

**Figure 1 cancers-13-04517-f001:**
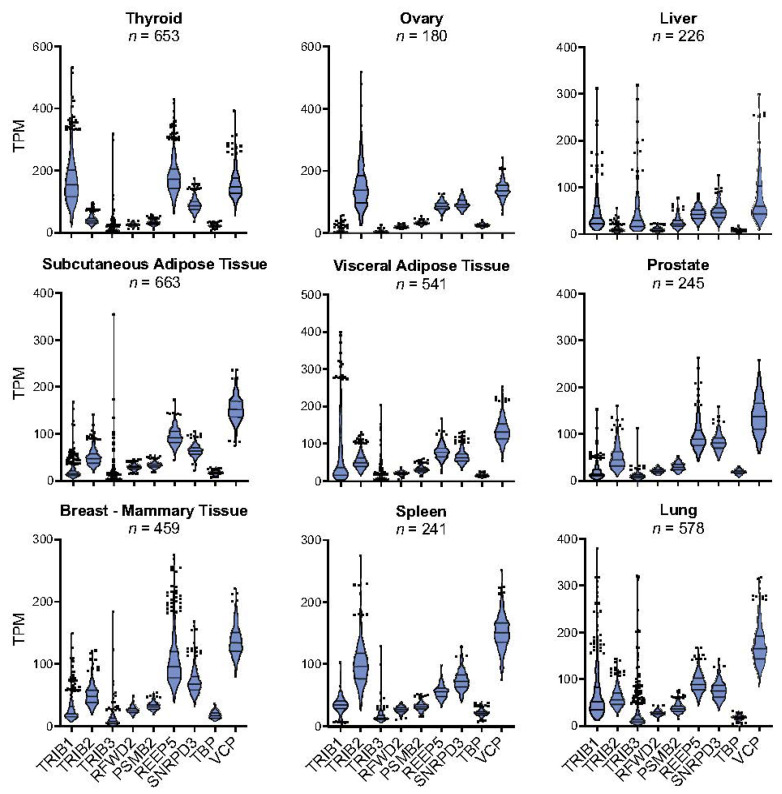
Violin plots of RNA-sequencing data from GTEx consortium (v8) [37] in a selection of tissues. Horizontal lines indicate median, 25th and 75th percentile values; black dots represent values >1.5 × the interquartile range below or above these values. TPM = transcripts per million. PSMB2 = proteasome subunit beta 2; REEP5 = receptor accessory protein 5; SNRPD3 = small nuclear ribonucleoprotein D3; TBP = TATA-box binding protein; VCP = valosin-containing protein.

**Figure 2 cancers-13-04517-f002:**
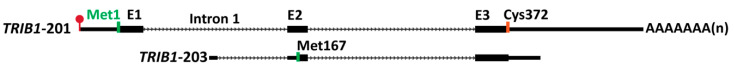
Example of annotated alternatively spliced *TRIB* transcript. Adapted from Ensembl and drawn to scale. N.B. Arrowed lines representing intronic sequences, which are removed from precursor *TRIB1*-201 (ENST00000311922) and *TRIB1*-203 (ENST00000520847) transcripts during splicing. Black rectangles and lines represent coding and non-coding sequences, respectively. Red dot and poly A tail highlight that the *TRIB-201* transcript is capped and polyadenylated. Met1 and Met167 indicate assumed translation initiation codons. E = exon.

**Figure 3 cancers-13-04517-f003:**
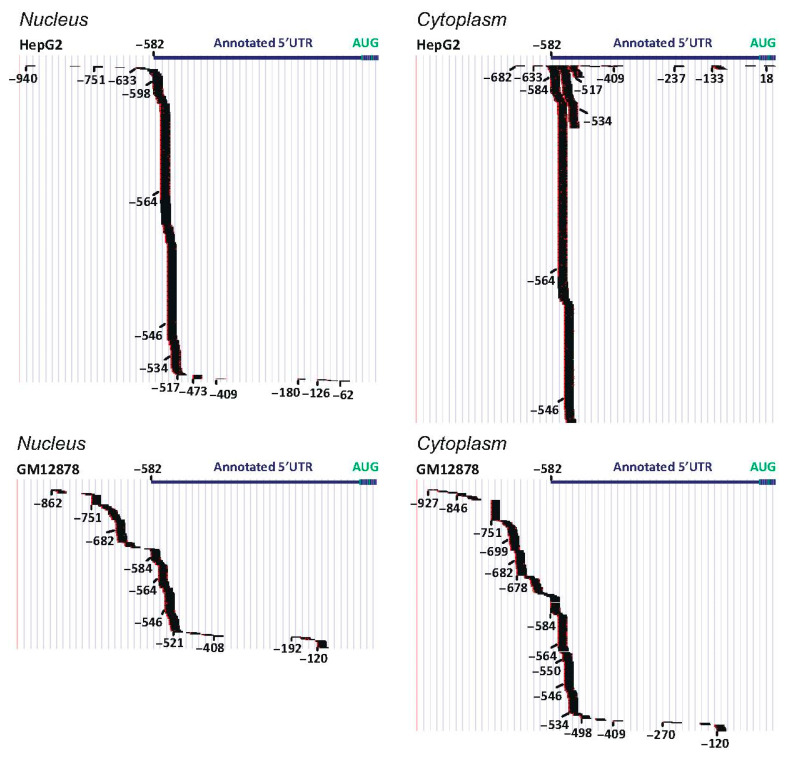
Initiation of *TRIB1* transcription from multiple sites within a core promoter. Visualisation of CAGE tags in the promoter region of *TRIB1*. Blue bar represents Ensembl-annotated *TRIB1*5′UTR. Numbering, in base pairs, is from the translation initiation codon (AUG, green). Each horizontal black bar represents a capped transcript: *n* = 1435 and 429 (nucleus) and 1670 and 534 (cytoplasm) in the 1kbp upstream of the translation initiation codon AUG in HepG2 and Epstein–Barr-virus-transformed lymphoblastoid cell line GM12878, respectively. Annotated sequence from the GRCh37/hg19 Assembly, drawn to scale.

**Figure 4 cancers-13-04517-f004:**
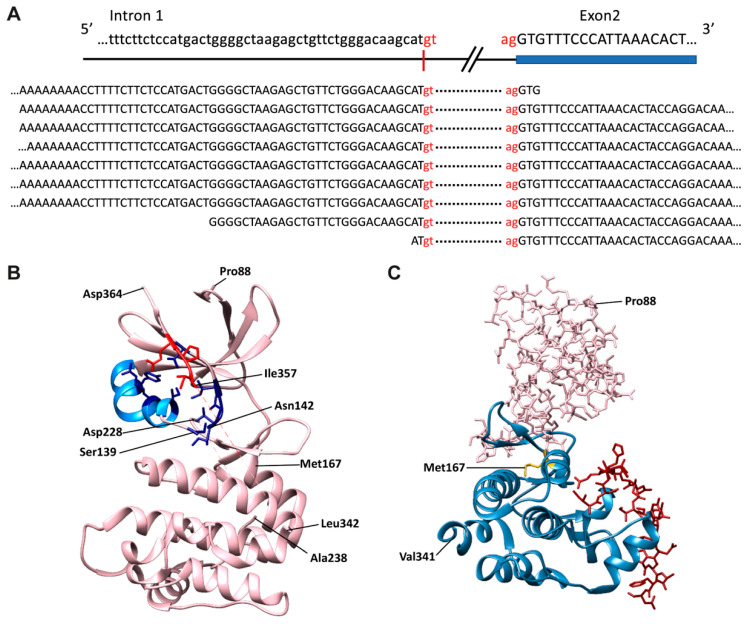
Analysis of putative *TRIB1*-203 gene-product. (**A**)*TRIB1*-203 transcripts detected in whole-cell total RNA extracted from human diencephalon, frontal cortex, occipital and parietal lobe samples. RNA-seq reads, downloaded from ENCODE, aligned against reference genome hg38. Nine *TRIB1*-203 transcripts were detected in these samples compared to 263 reads crossing the canonical exon 1 and 2 junctions. Nucleotides “gt” and “ag” define the splicing junction represented by the dotted line and are not present in the read sequence. (**B**) Schematic of structure for amino acids 84–365 of substrate-free TRIB1 (5CEM) [52]. Selected residues in αC-helix (depicted in blue) and β4-strand forming the COP1-binding peptide (red) binding pocket are depicted as sticks. Pink dotted lines: no electron density for amino acids 229–237 and 343–356. (**C**) Structure for amino acids 53–75 of C/EBPα (brown) bound to N- and C-terminally truncated TRIB1 [53]. Blue ribbons represent structures predicted to be encoded by the *TRIB1*-203 transcript.

**Figure 5 cancers-13-04517-f005:**
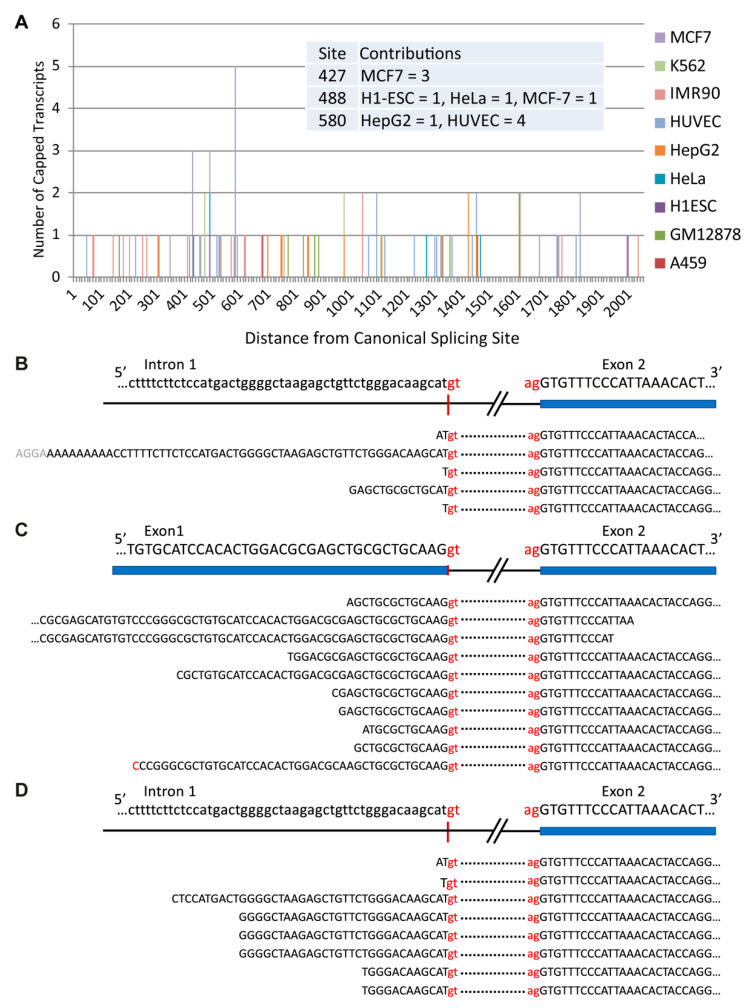
Distinguishing features of nuclear and cytoplasmic *TRIB1*-203 transcripts. (**A**) Capped transcripts in nuclear, poly A-enriched RNA in nine of ten examined ENCODE cell lines. Numbering (*x*-axis), nucleotides from beginning of intron 1. No CAGE tags were identified in the SK-N-SH cell line established from a bone marrow metastasis from a young female with neuroblastoma. CAGE tags (number, *y*-axis) mapping upstream of nucleotide 866 in intron 1 represent potential transcription start sites for the *TRIB1*-203 transcript. Corresponding number of (nuclear) CAGE tags mapping to the core promoter of *TRIB1*-201 range from 253 in the K562 cell line derived from an adult chronic myelogenous leukaemia patient in blast crisis to 2955 in the human lung fibroblast cell line IMR90. (**B**) RNA-seq reads for *TRIB1*-203 transcripts (*n* = 6 in total) detected in nuclear, poly A-enriched RNA of three (two replicates of each) ENCODE cell lines: HeLa, HUVEC and IMR90. N.B. only one read from each replicate is shown. (**C**) The first ten RNA-seq reads (*n* = 375) represent *TRIB1*-201 transcripts in nuclear, poly A-enriched RNA extracted from HepG2 cells. (**D**) RNA-seq reads for *TRIB1*-203 transcripts (*n* = 8 in total) detected in cytoplasmic, poly A-enriched RNA of the four specified ENCODE cell lines. Corresponding numbers for the *TRIB1*-201 transcript range from 86 for the K562 cell line to 1694 for the carcinomic human alveolar basal epithelial cell line.

**Figure 6 cancers-13-04517-f006:**
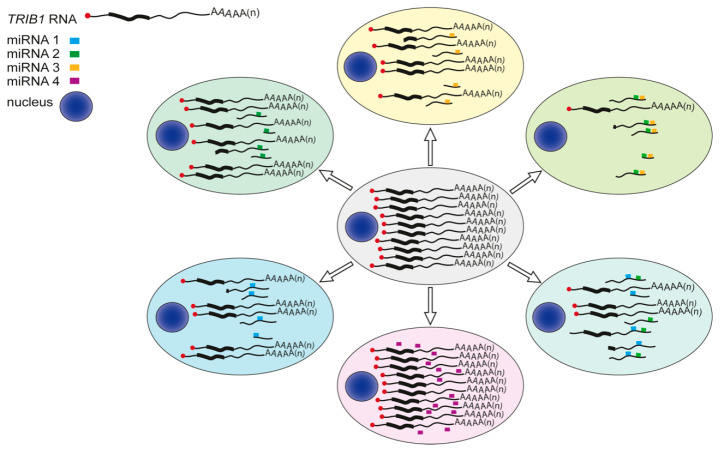
Cell-specific microRNA targeting of capped, polyadenylated *TRIB1* transcripts. Newly-synthesised, capped, polyadenylated *TRIB1* transcripts are subjected to microRNA (miRNA)-mediated de-adenylation and translational repression. Newly exported *TRIB1* transcript (centre cell) fate will vary on cell-type, physiological cues and miRNA compliment (coloured cells). The relatively long (~2000 nucleotides) 3′ untranslated region of the *TRIB1*-201 transcript contains potential binding sites for 1237 different miRNA [56]. Binding of miRNAs to closely-spaced sites (2 o’clock position) may facilitate cooperative behaviour leading to more de-adenylation and translational repression of *TRIB1*-201 transcripts than that achieved through combinational binding of miRNAs to target sites separated by >~60 nucleotides [57] (4 o’clock position).

**Table 1 cancers-13-04517-t001:** Characteristics of prostate cancer cell lines used by TRAIN consortium.

Cell Line	Characteristics	TRIB1 Protein Levels ^1^	COP1 Protein	Publications
PC3	PTEN^–^ [61]		Genome rearrangement	Shahrouzi et al., 2020 [59]
	Androgen-independent [62]	High	undetectable protein [63,64]	
	Bone metastasis			Niespolo et al., 2020 [56]
DU145	PTEN^+^ [65]			
	Androgen-independent [66]	NA	Yes [63,67]	Shahrouzi et al., 2020 [59]
	Brain metastasis			
LNCaP	PTEN^–^ [61]			
	Androgen-sensitive [62]	Low	Yes [63,64,67]	Niespolo et al., 2020 [56]
	Lymph node metastasis			
22Rv1	PTEN^+^ [68,69]			
	Androgen-sensitive [62]	Low	Yes, based on RNA ^2^	
	Prostate			

^1^ TRIB1 protein levels defined in a comparative manner based on experiments from Mashimaet al. [58]; NA = Not Available: not included in comparative analysis. ^2^ Data from Cancer Cell Line Encyclopaedia.

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
