# Peer review of "Musings from the Tribbles Research and Innovation Network"

_cancers, 2021, doi:10.3390/cancers13184517_

Round 1
Reviewer 1 Report
The authors present a review of the state of the art of research into the function and pathological role of the TRIB family of of protein pseudokinases, as studied by members of the TRAIN Consortium. I have very little to say, other than a couple of very minor suggestions:
- I did find myself looking fora very simple summary statement of the function of TRIB(s) in mammalian cells, similar to that supplied to Drosophila. The information was there, but scattered across several sections and often written to descriibe as the observation made during the experiment rather than conclusions derived from it. For those not familiar with TRIB biology, or even what a pseudokinase is, a little more basic information about the proteins may be useful. I did find such as statement for the Drosophila protein but would have found this useful at the start of the article to put the subsequent information about the proteins in context. In trying to describe both the protein and the work of the Consortium together, I think the focus on the protein was lost. Alternatively a summary figure could be added and referred to.
- The authors refer to a TRAIN toolbox, some aspects of which they imply are publicly available, bu they have not added a link to the TRAIN website which I assume would be the access point for people wishing to use these resources
Author Response
We thank the reviewer for their critique of the Commentary and have addressed the two minor revisions requested, increasing the cohesiveness and readability of the commentary.
Reviewer 1 . The authors present a review of the state of the art of research into the function and pathological role of the TRIB family of protein pseudokinases, as studied by members of the TRAIN Consortium. I have very little to say, other than a couple of very minor suggestions:
- I did find myself looking fora very simple summary statement of the function of TRIB(s) in mammalian cells, similar to that supplied to Drosophila. The information was there, but scattered across several sections and often written to describe as the observation made during the experiment rather than conclusions derived from it. For those not familiar with TRIB biology, or even what a pseudokinase is, a little more basic information about the proteins may be useful. I did find such as statement for the Drosophila protein but would have found this useful at the start of the article to put the subsequent information about the proteins in context. In trying to describe both the protein and the work of the Consortium together, I think the focus on the protein was lost. Alternatively a summary figure could be added and referred to.
Response: we thank the reviewer for their suggestions and have added more basic information upfront about the potential functions of mammalian TRIB proteins in several places:
a). In the Introduction, we have added ‘while the inclusion of the reported TRIB1 protein structures highlights how the central kinase-like domain of the mammalian TRIB proteins may bind a specific, yet diverse, range of proteins. Notably, the structures also provide insights into how these pseudokinases bind the E3 ubiquitin ligase COP1 via a C-terminal motif to promote proteasomal-mediated degradation of TRIB-captured substrates.’
b). In the Introduction we have highlighted the function of TRIBs in mammalian cells relates to it binding to a diverse set of proteins via its central kinase-like domain (see a, above). In line 58, we have also changed the sentence slightly (as highlighted in red) ‘and share research tools for studying the complexities of the cell-specific activities of TRIB proteins in common potentially obesity-related diseases, including prostate [7] and breast [8] cancer.’
c). added to lines 65-67 the following ‘including the design of small molecules that selectively modulate physical contacts between the central kinase-like domain of TRIB proteins with a specific protein or subset of proteins.'
d). changed the first sentence of the paragraph describing the activities of mammalian TRIB proteins (lines 79-80). It now reads: ‘Four discoveries hinting at the potential cell-specific roles of mammalian TRIB proteins in regulating a diverse range of cell signalling activities, and the ensuing collaborations arising from them, were pivotal in inspiring and creating the TRAIN.
e). In the Conclusion (lines 344-361). We have re-iterated why the TRIB1 proteins have diverse cellular roles and concluded how the TRIB/COP1 axis could be targeted to modulate selective aspects of these TRIB activities.
- The authors refer to a TRAIN toolbox, some aspects of which they imply are publicly available, bu they have not added a link to the TRAIN website which I assume would be the access point for people wishing to use these resources
Response: This is addressed on line 312.

Reviewer 2 Report
The manuscript by Cantos et al 2021 provides a commentary on the historical and active research program focused upon Tribbles proteins and their cellular influence upon tumorigenesis. The authors provide an interesting historic perspective that included key discoveries that helped improve understanding of the function of Tribbles proteins family members. The authors then provide insight into different protein isoforms and the intricacies of their differential message expressions. Further details covering genomic and structural data is included. Lastly, the authors provide details highlighting and publicising their association with the Tribbles Research and Innovation Network (TRAIN). Although this is a somewhat personalized review of elements of this research area and the TRAIN network, the commentary/review is of interest to readers in the field. I, therefore, have no reservations about recommending publication without revision.
Author Response
Reviewer 2. The manuscript by Cantos et al 2021 provides a commentary on the historical and active research program focused upon Tribbles proteins and their cellular influence upon tumorigenesis. The authors provide an interesting historic perspective that included key discoveries that helped improve understanding of the function of Tribbles proteins family members. The authors then provide insight into different protein isoforms and the intricacies of their differential message expressions. Further details covering genomic and structural data is included. Lastly, the authors provide details highlighting and publicising their association with the Tribbles Research and Innovation Network (TRAIN). Although this is a somewhat personalized review of elements of this research area and the TRAIN network, the commentary/review is of interest to readers in the field. I, therefore, have no reservations about recommending publication without revision.
Response: Thank you. Minor typos were found and corrected.